# Efficacy of Vitamin D Supplementation in Addition to Aerobic Exercise Training in Obese Women with Perceived Myalgia: A Single-Blinded Randomized Controlled Clinical Trial

**DOI:** 10.3390/nu13061819

**Published:** 2021-05-27

**Authors:** Heba Ahmed Ali Abdeen, David Rodriguez-Sanz, Mahmoud Ewidea, Dina Mohamed Ali Al-Hamaky, Marwa Abd El-Rahman Mohamed, Ahmed Ebrahim Elerian

**Affiliations:** 1Department of Physical Therapy for Cardiovascular/Respiratory Disorder and Geriatrics, Faculty of Physical Therapy, Cairo University, Ad Doqi, Giza District, Giza Governorate 11432, Egypt; 2Faculty of Nursing, Physical Therapy and Podiatry, Universidad Complutense de Madrid, 28040 Madrid, Spain; davidrodriguezsanz@gmail.com; 3Department of Basic Science for Physical Therapy, Faculty of Physical Therapy, Kafr Elshiekh University, Kafr Elsheikh Government 33511, Egypt; Mahmoud_auida2014@pt.kfs.edu.eg; 4Department of Physical Therapy for Musculoskeletal Disorders and Its Surgery, Faculty of Physical Therapy, Cairo University, Ad Doqi, Giza District, Giza Governorate 11432, Egypt; dinaali2008@cu.edu.eg; 5Department of Physical Therapy for Women Health, Faculty of Physical Therapy, Cairo University, Ad Doqi, Giza District, Giza Governorate 11432, Egypt; Marwa.abdelrahman@cu.edu.eg; 6Department of Basic Science for Physical Therapy, Faculty of Physical Therapy, Cairo University, Ad Doqi, Giza District, Giza Governorate 11432, Egypt

**Keywords:** obesity, myalgia, vitamin D, aerobic interval training, functional capacity, quality of life

## Abstract

Obese women were more susceptible to myalgia because of their significantly lower vitamin D concentrations; the present study investigated the efficacy of vitamin D in addition to an aerobic interval training in the management of obese women with myalgia. Forty-five obese women with vitamin D deficiency and myalgia (30 to 40 years old) were assigned randomly into three equal groups. Group A received an aerobic interval training with vitamin D supplementation, Group B received vitamin D supplementation only, and Group C received aerobic interval training only; participants in all groups were on calorie deficient diets. The study outcomes were the Visual Analog Scale (VAS) for Pain Evaluation, serum vitamin D level, and Cooper 12-Minute Walk Test for Functional Capacity Evaluation, while the Short-Form Health Survey (SF) was used for assessment of quality of life. We detected a significant improvement in pain intensity level, serum vitamin D level, and quality of life in all groups with significant difference between Group A and groups B and C. We also detected a significant improvement in functional capacity in groups A and C, with no significant change in Group B. Aerobic interval training with vitamin D supplementation was more effective for the management of obese women with perceived myalgia.

## 1. Introduction

Vitamin D deficiency is common within the female adult population and can be linked with reversible myalgia. Vitamin D deficiency is predominant among patients with hyperlipidemia but is yet more likely among patients with statin-associated myalgia [1]. It is probable that vitamin D deficiency potentiates statin-induced myalgia or cause drug-unrelated myalgia in a subset of statin-treated patients [2].

Reduction of vitamin D3 bioavailability from dietary and cutaneous sources is the main cause of vitamin D insufficiency associated with obesity due to its deposition in body fat compartments [2]. 

The main cause of vitamin D insufficiency associated with obesity in females remains unclear, but it may be due to variation in vitamin D consumption, less exposure to the sun, reduced skin biosynthesis, hereditary variety, or inherent features related to obesity, such as adipose tissue’s role in increasing the sequestering of fat-soluble vitamin D, or due to sensitivity of vitamin D serum concentrations to weight loss changes [3]. Serum 25-hydroxyvitamin D [25(OH) D], which is considered the main clinical indicator of vitamin D status, had been shown to be inversely correlated with obesity [4].

Several studies have recorded constant, nonspecific musculoskeletal pain in patients secondary to severe hypovitaminosis D and its serious implications on bone health and quality of life [5,6]. Despite plentiful sunshine, rickets rates are highest in the Middle East and Africa. This is likely because of constrained sun exposure due to cultural practices and prolonged breastfeeding without vitamin D supplementation in the Middle East, as well as by dark skin color and calcium deficiency, rather than because of vitamin D insufficiency. Nevertheless, both provinces have a high incidence of hypovitaminosis D [7].

Obesity is considered a major risk factor for insulin resistance, impaired glucose tolerance, dyslipidemia, dysfunctional bleeding, and cardiovascular diseases, as well as increasing the risk factor for developing the Poly Cystic Ovary (PCO) [8]. Several studies had focused on the significant relationship between pain and obesity, also on the effect of obesity on soft-tissue structures, such as muscle, tendons, fascia, and cartilage [9,10,11].

Weight loss could reduce the risk factors for type 2 diabetes, cardiovascular disease, and endocrine and reproductive parameters in obese female subjects [12]. Weight loss also helps obese chronic pain patients [13].

Therefore, lifestyle factors, as well as diet and physical activity targeting weight reduction, have been demonstrated to affect the pain condition in obese subjects [5,6]. It is well established that exercise and/or engaging in a hypocaloric diet [14,15,16] improves body composition in overweight and obese individuals by reducing systemic insulin levels, which in turn, reduces adipogenicity and lipogenic mechanisms [17,18]. Beyond diet and exercise interventions, supplementing the diet with vitamin D could enhance weight loss and modify the metabolic alterations associated with obesity, therefore modifying the obesity-disease relationship [18,19].

Additionally, many studies revealed that aerobic exercises have moderate beneficial effects on physical function [20,21], tender points, and pain in patients with fibromyalgia [22]. Exercise with vitamin D3 supplementation can also improve muscle strength in elderly adults [23].The effectiveness of vitamin D supplementation alone or in combination with exercise on obese patients with fibromyalgia, in addition to hypocaloric balanced diets, has also been investigated [24].

The findings of this study will help understand the combined effects of aerobic interval training and vitamin D supplementation on functional capacity and perceived myalgia in middle-aged obese women. These data will help guide physical therapy toward better results through decreasing the time needed to perform the activities of daily living independently and making the patient an active member of society.

## 2. Materials and Methods

Design and setting: A single-blinded randomized controlled clinical trial. 

The investigation was done according to the 1975 Declaration principles of Helsinki. The study had been prospectively registered at the Clinical Trials Registry (Clinical Trials.gov, NCT04319289) after its approval by the Research Ethical Committee of the Faculty of Physical Therapy, Cairo University (3/12/2019, P. T. REC/012/002211). All patients provided written informed consent for their information and anonymous data to be published. This randomized controlled trial includes a completed CONSORT flow chart and is conducted in accordance with CONSORT guidelines [25].

Participants: Sixty obese women suffered from nonspecific muscle pain and were referred to physical therapy by a specialized orthopedist. The patients were enrolled in this study if they were 30–40 years old, their body mass index (BMI) was within the class I obesity category (BMI, 30.0 to 34.9 kg/m^2^) [26], they perceived myalgia, with nonspecific muscle pain for more than 2 months, and had vitamin D deficiency (serum 25(OH) D levels <30 ng/mL and <10 ng/mL). Fifteen women were excluded after referral due to the presence of one of the following ailments: unstable cardiovascular disease, cardiopulmonary disorder, metabolic diseases, rheumatic disease, and any bone diseases or musculoskeletal disorders that affect vitamin D level and the patients’ ability to perform the therapeutic exercises, as well as any patient under any medication that might affect vitamin D levels, such as steroids and anticonvulsants drugs (Figure 1). The study was conducted in the outpatient clinic of the Faculty of Physical Therapy, Cairo University, between January 2020 and May 2020.

Sample size and Randomization: (G power program version 3.1, Heinrich-Heine-University, Düsseldorf, Germany) for the one-tailed test was used to calculate the sample size. The previous studies addressing vitamin D3 supplementation and resistance exercise training on musculoskeletal health were used to determine the effect size for the sample size calculation. According to previous studies, 13 patients in each group was estimated to be the minimum sample size needed to achieve a power of 90% to detect an effect size of 0.87 in the outcome variables [27,28], assuming a type II error (Alfa = 0.05). To account for dropouts during the study, the sample size increased by 15%:20 patients in each group. The remaining 45 women with nonspecific muscle pain and vitamin D deficiency participated in the study and were randomly assigned to three groups. Group A (*n* = 15) received a combination of an aerobic interval training exercise program and vitamin D supplementation (cholecalciferol 50,000 IU/week) [29]. Group B (*n* = 15) received vitamin D supplementation only (cholecalciferol 50,000 IU/week). Group C (*n* = 15) received an interval training exercise program only (Figure 1), while each participant in the three groups followed an individualized low-fat, balanced diet. 

Sealed opaque envelopes were used to confirm a single-blinded random assignment in which the patients involved in the study were unaware of assignment to treatment groups.

Outcome measures: All study variables were evaluated before the subject’s allocation to groups and immediately after the end of the study (12 weeks) for all groups.
Laboratory analyses: The BioPlex^®^ 2200 System (Hitachi, model 704,902, Japan) was used for measuring serum vitamin D level (25OHD), Fasting (12 h) venous blood samples (10 mL) were collected in the morning at 9 a.m. by a specialist technician from the antecubital vein in Cairo University Hospital clinical lab using an anticoagulant free tube (EDTA K3) [30].Functional capacity: The Cooper 12-Minute Walk Test (12MWT) was used to evaluate the functional capacity efficiency for each patient in each group [31]. Each patient was asked to warm up for 10 min, then the command “START” was given, a stopwatch was started, and the patient began running. The patient was informed of how much time she had remaining at the end of each completed lap. At 12 min, a whistle was blown, the patient stopped, and the distance was recorded [32].Intensity of perceived myalgia (nonspecific muscle pain): An Arabic version of a 10-cm Visual Analog Scale (VAS) was used to evaluate the patients’ pain [33,34], with 0 representing no pain and 10 representing the worst pain imaginable [35]. Each patient in each group was asked to rate their pain at rest and on motion in the previous week using a 0–10 cm VAS just before and after the study duration. The measure was taken by a ruler from the right zero point (in Arabic version) to the mark the patient scored in millimeters. Assessments of the minimal clinically important difference for the VAS indicate that a change of 1.5 to 2.0 cm (15–20%) is required to detect improvement [36].Quality of Life (QOL): The short-form health survey (SF-12) was used to compare health status between groups of patients and to identify predictors of health status and QOL. The SF-12 contains 12 items. All SF-12 items came from the SF-36. This includes eight dimensions: physical functioning, role limitations due to physical health problems, bodily pain, general health, vitality, social functioning, role limitations due to emotional problems, and mental health [37]. The SF-12 is a shorter version of the SF-36 and uses only 12 questions to measure functional health and wellbeing from the patient’s perspective. The original objective was to develop a short, generic health status measure that reproduces the two summary scores of the SF-36, i.e., the physical component summary (PCS) score and the mental component summary (MCS) score [38].

The scoring system used for scoring of the SF-12 questionnaire was based on the scoring oftheSF-12 from Hanmer [39].

Scoring of SF-12: The SF-12 contains a mixture of positively (higher scores indicate better health) and negatively worded response scales, so some items need to be recorded prior to scoring. The scale scores are calculated by summing responses across scale items and then transforming these raw scores to a 0–100 scale. TheSF-12 can also be used to derive two aggregate summary measures: the PCS and the MCS. Summary scores are calculated by summing factor-weighted scores across all eight subscales [40].
5.Height evaluation: Stable stadiometer for mobile height measurement (Seca 217) was used for the evaluation of each patient’s height (cm).6.Body mass index evaluation: An InBody 270 body composition analyzer was used to calculate body mass index (BMI) [41].7.Food frequency questionnaire (FFQ): Individual adherence to the prescribed diet was evaluated every 2 weeks by a self-recorded food frequency questionnaire. The evaluation of the diet followed by the participants in this study was done using FFQs, through inquiring about the rate of recurrence of ingested foodstuff or precise food groups over a certain duration of time [42]. It is appropriate for mature people to congregate information on a wide variety of foods and can be designed to be shorter and concentrate on foods rich in a specific nutrient or on a specific group of foods [43]. FFQ, which contained the six food groups (fruit, vegetables, starch, milk, meat, and fat) in specific units to be measured in gram, was used to measure the adherence of each subject in each group, then the adherence in each group was calculated. Additionally, the relation between the groups and level of adherence was calculated using 3 × 2 Chi-Square.

### Interventions

Low-calorie balanced diet: Each participant in the three groups of the study consumed an individually tailored low-calorie balanced diet that created a caloric deficit of 500 Kcal/day with the daily energy requirement, as well as macronutrient intake of the balanced healthy diet. A low-calorie, balanced diet was designed based on in personal characteristics of each subject and with the goal of 500 Kcal daily caloric restriction relative to the total energy requirements (TEE) calculated by the Mifflin formula [29]. This prescribed individualized regimen was reviewed by the nutritionist every 14 days. A well-balanced low-calorie diet program that involved three meals was designed according to the recommended dietary intake (RDI), while the macronutrients were considered in the diet, as follows: fat 20–30%, protein 10–15%, and carbohydrates 55–65% of TEE, and rich in dietary fiber, as well as fruit and vegetables [29,44].

Aerobic interval training: aerobic exercises were performed by each patient in group A and using a stationary bicycle ergometer (Monark Rehab trainer: 88E). Patients in groups A and C were instructed to perform aerobic interval training at 70% maxHR for 5 min and then decrease activity to 50% for 5 min, then repeat for the length of the activity [45]. Before starting training, the maximum heart rate was calculated using the following equation: max HR = 220-age. 

The duration of each training session was divided into three phases. The first phase was the warm-up phase that lasted 5–10 min in a slow progression exercise to avoid cardiovascular risk and musculoskeletal injury, while the second phase was the active phase of exercise for 40 min, and the third phase was the cool-down phase that lasted 10 min, in gradually reduced intensity and speed until reaching the resting HR with the frequency of training 3 days/week, for 12 weeks.

Vitamin D supplementation: The subjects in groups A and B received supplementation. One capsule containing cholecalciferol 50,000 IU; (Biodal^®^ 50 000 IU cholecalciferol; HAYAT PHARMACEUTICAL INDUSTRIES, Amman, Jordan) was taken every week [23,29].

Statistical analyses: Prior to the final statistical analyses, the raw data were monitored and examined for normality and homogeneity. The data were tested for normality using the Shapiro–Wilk test. The data were distributed normally after the removal of outliers detected using box and whiskers plots [46]. Levene’s test was used for testing the homogeneity of variance and revealed that the data were normally distributed and parametric analysis (*p* > 0.05). Statistical SPSS Package program version 25 for Windows (SPSS, Inc., Chicago, IL, USA) was used to apply a multiple imputation model for the missing data (those outlier data identified for removal by the box and whiskers plot). This technique allows for the uncertainty about the missing data by creating several different plausible imputed data sets and appropriately combining results obtained from each of them [47]; descriptive statistics were used to compare age, BMI, VAS, vitamin D, PCS, MCS, and 12MWT variables in the current study. A mixed design multivariate analysis of variance (MANOVA) was used to compare the tested variables of interest at different tested groups and measuring periods. The Bonferroni correction test was used to compare between pairs of groups for post-treatment of the tested variables, in which the F-value was significant from the MANOVA test. *p*-values ≤ 0.05 were considered statistically significant.

## 3. Results

The basic characteristics of the participating patients in three groups are given in Table 1. There were no significant differences in age (*p* = 0.82; *p* > 0.05) and BMI (*p* = 0.98; *p* > 0.05) among the groups. There was no significant decrease in VAS and BMI at the post-treatment measurement in groups A, B, and C (*p* < 0.0001). There were no significant differences (*p* = 0.68) in mean VAS and BMI values at the pre-treatment evaluation among the three groups. However, there were significant differences (*p* = 0.001) in the mean values of post-VAS among the three groups. The greatest decrease in VAS was in Group A, followed by group B and group C. The greatest decrease in BMI was also in Group A, followed by group C and Group B (Table 2).

Regarding the vitamin D level, we detected a significant difference at post-treatment compared to pre-treatment in groups A, B, and C (*p* = 0.0001; *p* < 0.05). The greatest percentage improvement was detected in group A, with no significant differences in mean values of pre-treatment vitamin D between the three groups (*p* = 0.119; *p* > 0.05). However, there were significant differences in the mean values of post-treatment vitamin D among the three groups (*p* = 0.0001; *p* < 0.05). Again, this increase was greatest in Group A, followed by groups B and C (Table 2 and Table 3).

Regarding the quality of life assessment, the present study revealed that the PCS post-treatment significantly increased compared with that of pre-treatment (*p* = 0.0001) within group A, but there were no significant differences within B and C (*p* = 0.218 and 0.232, respectively).However, for MCS, the post-treatment value significantly increased compared to pre-treatment within groups A and C (*p* = 0.0001 and 0.047, respectively), while there was no significant difference within group B ( *p*= 0.272). PCS and MCS pre-treatment comparisons showed that there were no significant differences between the three groups (*p* = 0.330 and 0.507, respectively), with significant differences in the mean values of PCS and MCS post-treatment comparison between the three groups (*p* = 0.001) (Table 2). Additionally, the Bonferroni test and mean difference of PCS showed that Group A had the highest PCS and MCS values, with significant differences between groups: Group A vs. Group B and Group A vs. Group C (*p* = 0.001 and 0.023, respectively) PCS while for MCS the *p*-value equal 0.001 and 0.018(*p* < 0.05) respectively, and no significant difference between group B versus group C while the *p*-value equal 0.860 for PCS and 0.7 for MCS (*p* > 0.05).

Regarding the functional capacity assessment, there was a significant increase of 12MWT score at post-treatment with comparison to pre-treatment within groups A and group C (*p* = 0.0001 and 0.014, respectively), while for group B, no significant difference was detected (*p* = 0.319). Moreover, the pre-treatment analysis for 12 MWT scores revealed no significant differences between the three groups of the study (*p* = 0.161), but the post-treatment analysis for 12 MWT scores revealed significant differences between the three groups of the study (*p* = 0.001). The Bonferroni test and mean difference of 12MWT between pairwise of the groups showed that group A had the highest scores in 12MWT values, with significant differences between group A versus group B (*p* = 0.0001) and also revealed no significant differences in 12MWT between group A versus group C and group B versus group C (*p* = 0.159 and 0.069, respectively) (Table 3).

A Chi-square test of independence was performed to examine the relationship between the groups and adherence to the prescribed diet. The relationship between these variables was not significant, X2 (1, *N* = 45) = 1.4516, *p* = 0.483934. Therefore, the level of adherence to the prescribed diet did not affect the relationship between the groups (Table 4).

## 4. Discussion

This study was designed to investigate the combined effects of aerobic interval training and vitamin D supplementation on functional capacity and perceived myalgia in middle-aged obese women. This was in line with a previous study reporting that vitamin D deficiency was more common in women than men, especially women with BMI greater than 25, so the included women with mean age 35.07 ± 2.4 years and BMI 34.18 ± 3.41 kg/m^2^ represent a particularly high-risk situation [48]. 

Our results show significant improvement in the BMI in the three groups, with the greatest improvement in group A. This improvement in all groups of the study could be mainly linked to the effect of a low-calorie balanced diet that had a positive effect alone on the weight and BMI [49,50]. Additionally, the more positive effect in group A could be linked to the effect of aerobic exercise, as noted in previous studies [51,52,53], as well as because of vitamin D supplementation in addition to aerobic exercise. A previous study has reported that when vitamin D intake is combined with exercise training, this results in greater improvements in body composition, abdominal fat, blood lipid, and insulin resistance index than situations with one single treatment [54]. Additionally, here we found that group B had the lowest improvement in BMI. The improvement in this group might be to the effect of a balanced low-calorie diet only as the vitamin D supplementation alone had no significant effect on BMI; this was in line with the recent systemic review that reported that despite increased serum 25(OH) D levels, the obesity indices (BMI, Waist Circumference and Waist hip Ratio) did not improve [55].

Our results indicated a significant increase in serum vitamin D in all groups, with the greatest increase in group A. The greater improvement in Group A than Group B could be due to the effect of vitamin D supplementation, as reported by Pilz et al. [56], who detected positive effects of vitamin D supplementation on serum 25-hydroxyvitamin D concentrations, while the improvement of serum vitamin D in Group C—which received the aerobic exercise only—could be correlated with increased vitamin D receptors in muscle, pancreas, and adipose tissue [57].This result is consistent with the results of Gholamalizadeh et al. [58] who concluded that serum vitamin D levels increased significantly at the end of the training program in overweight and obese adolescents. Our results also align with the previous report of The National Health and Nutrition Examination Survey (NHANES III), which reported that physical activity could positively affect serum 25(OH) D3 levels due to enhanced vitamin D metabolism [59].

We detected a significant decrease in pain intensity according to VAS score in all groups, with a less significant decrease in Group C relative to groups A and B. This enhancement might be attributed to interval training exercises, which increases the blood flow and nutrients to the soft tissues in the body, improving the healing process and reducing the stiffness that results in musculoskeletal pain, in addition to the chemical effect of aerobic interval exercise through increasing the body’s production of endorphins, often referred to as a natural opiate that results in pain relief [60]. Additionally, vitamin D supplementation can decrease pain scores by inducing a decrease in the levels of inflammatory and pain-related cytokines in plasma, such as prostaglandin and leukotrieneB4 [61].

This improvement of pain level in groups A and B could be because of improvement of vitamin D level as well as to aerobic exercise, as the improvement in vitamin D status in obese subjects with vitamin D deficiency resulted in weight, fat mass, and monocyte chemoattractant protein-1 (MCP-1) decrease. Therefore, weight loss and vitamin D supplementation likely act synergistically to reduce levels of meta-inflammation [29].

Our findings are consistent with those of another study that concluded that aerobic interval exercise appears to be a feasible rehabilitation approach in patients with nonspecific musculoskeletal pain [62]. Additionally, another study by Yilmaz et al [63], who endorsed the impact of vitamin D supplementation on the advancement of pain and quality of life on patients endured from persistent nonspecific musculoskeletal pain, the mechanism of this improvement was outlined in a previous study that showed that vitamin D supplementation resulted in reduced levels of inflammatory and pain-related cytokines in plasma [61].

Concerning the functional capacity, in the current study, the Cooper 12-Minute Walk Test (12MWT) was used to assess the functional capacity for all groups pre- and post-study; improvement occurs when there is an increase in the distance walked in 12 min. We detected a significant increase in groups A and C, with no significant difference in the distance walked when comparing these groups. In group B, there was no significant increase, but with significant differences relative to groups A and C. The improvement in both groups A and C may be attributed to the physiological effect of an interval training exercise on the body [62].

This functional capacity improvement was linked to an increase of VO_2_max percentage that depending on the age and fitness level of the subjects as well as the duration and intensity of the interval aerobic training [64]. Regarding the effects of aerobic exercise on functional capacity, the results of the present study agreed with multiple published studies [65,66,67]. Regarding the effect of vitamin D supplementation on physical function and functional capacity, our results are in line with many previous studies [61,68].

Regarding the quality of life: A short-form health survey (SF-12) was used to assess the health status of the patients in all groups pre- and post-study in this study. The results of this study showed that there was a significant increase in both PCS and MCS of SF-12 in all groups, with a significant difference between Group A and groups B and C.

The results of the present study agree with those of Isaksen et al. [69], who conducted a study to determine the effects of aerobic interval training on measures of anxiety, depression, and quality of life in patients with ischemic heart failure and revealed that interval training exercises showed significant improvements in several SF-36 subscores at 12 weeks. Therefore, anaerobic interval training program resulted in significant improvements in several measures of quality of life. 

In accordance with the present study, a study was conducted to investigate the effect of vitamin D supplementation on quality of life in women with type 2 diabetes. Fifty women with type 2 diabetes were enrolled into weekly vitamin D supplementation (ergocalciferol, 50,000 IU) for 6 months. Vitamin D supplementation induced a significant improvement in both components of the SF-12 survey. The study concluded that vitamin D supplementation could improve the health status of T2DM women [70].

The results of the present study contradict the finding of Westra et al. [71], who reported that 6 months of vitamin D supplementation did not improve health-related quality of life (using the SF-36). This might be due to the difference in the inclusive criteria of participants, as the Westra et al. study was performed on non-vitamin D deficient people with type 2 DM.

Our study has some limitations. One of these limitations was the investigation of the effect of adding the aerobic interval exercise to vitamin D supplementation with low balanced macronutrient diet only, without a true control or placebo group. Therefore, the results of the present study should be confirmed by other studies with different types of diet, such as a high protein diet or a Mediterranean diet using a true control or placebo group. Additionally, the present study evaluated the short-term effects following 12 weeks of follow-up. Future studies are recommended to investigate the long-term effect of aerobic interval exercise accompanied with vitamin D supplementation. Moreover, the present study did not examine the secondary hypercalcemia resulting from obesity, so future studies are warranted to investigate the effect of adding different exercises to vitamin D supplementation, including calcium level monitoring. 

## 5. Conclusions

Vitamin D supplementation combined with aerobic exercise in addition to a well-balanced hypocaloric diet could provide a novel protocol to improve the nonspecific musculoskeletal pain, functional capacity, and the quality of life in obese women with perceived myalgia.

## Figures and Tables

**Figure 1 nutrients-13-01819-f001:**
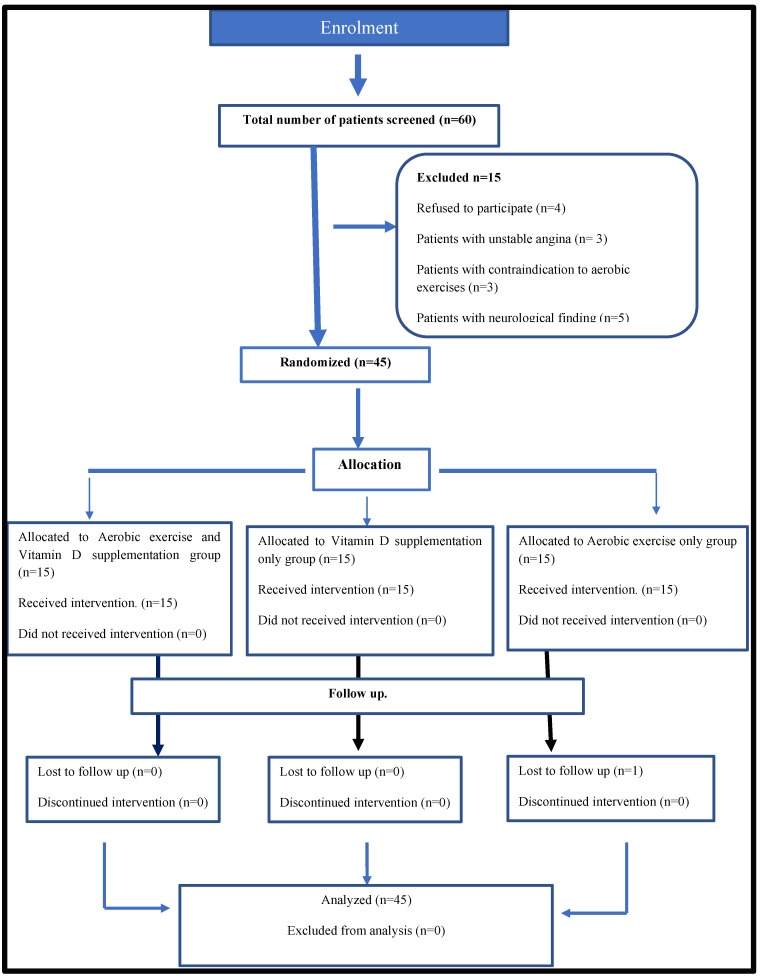
Flow chart in accordance with the CONSORT statement.

**Table 1 nutrients-13-01819-t001:** Basic characteristics of participated patients of groups A, B, and C.

Variable	Group A	Group B	Group C	*p*-Value
Age (Year)	34.8 ± 2.64	35.01 ± 2.39	35.4 ± 2.69	0.83
BMI (kg/m^2^)	34.17 ± 1.51	34.41 ± 1.85	33.96 ± 1.25	0.98

*p*-value: probability value.

**Table 2 nutrients-13-01819-t002:** Variables comparison within and among groups of the study.

Variables		Groups (Mean ± SD)			*p*-Value
		Group A	Group B	Group C	
VAS	Pre-treatment	6.1 ± 0.89	6.3 ± 0.7	6.4 ± 0.8	0.68
	Post-treatment	2.75 ± 1.25	3.03 ± 1.25	4.37 ± 0.85	0.001 *
	Improvement %	55.4%	52.07%	31.01%	
	*p*-value	0.0001 *	0.0001 *	0.0001 *	
VitD	Pre-treatment	17.10 ± 1.16	15.90 ± 1.67	16.36 ± 1.71	0.119
	Post-treatment	23.91 ± 4.29	19.87 ± 1.94	17.58 ± 2.29	0.0001 *
	Mean difference	6.81	3.97	1.22	
	Improvement %	39.82%	24.97%	10.75%	
	*p*-value	0.0001 *	0.0001 *	0.01 *	
PCS	Pre-treatment	45.60 ± 4.46	43.93 ± 3.59	45.93 ± 3.53	0.330
	Post-treatment	51.87 ± 4.24	45.87 ± 4.73	47.53 ± 3.54	0.001 *
	Mean difference	6.27	1.94	1.60	
	Improvement %	13.75%	4.42%	3.48%	
	*p*-value	0.0001 *	0.218	0.232	
MCS	Pre-treatment	46.20 ± 3.96	44.60 ± 3.86	45.60 ± 3.43	0.507
	Post-treatment	51.53 ± 2.99	46.33 ± 4.57	47.87 ± 2.44	0.001 *
	Mean difference	5.33	1.73	2.27	
	Improvement %	11.54%	3.88%	4.98%	
	*p*-value	0.0001 *	0.272	0.047 *	
12MWT	Pre-treatment	1627.20 ± 68.05	1575.13 ± 72.67	1615.00 ± 87.01	0.161
	Post-treatment	1758.00 ± 102.05	1604.93 ± 87.60	1687.93 ± 98.88	0.0001 *
	Mean difference	130.8	29.8	72.93	
	Improvement %	8.04%	1.89%	4.52%	
	*p*-value	0.0001 *	0.319	0.041 *	
BMI (kg/m^2^)	Pre-treatment	34.17 ± 1.51	34.41 ± 1.85	33.96 ± 1.25	0.98
	Post-treatment	27.75 ± 1.25	29.03 ± 1.25	27.91	0.001 *
	Improvement %	53.1%	41.98%	50.6%	
	*p*-value	0.0001 *	0.0001 *	0.0001 *	

SD: standard deviation *p*-value: probability value * Significant (*p* < 0.05).

**Table 3 nutrients-13-01819-t003:** Post-hoc test between pairwise of groups A, B, and C.

Variables		Post-hoc (Bonferroni Test)		
		Group A and Group B	Group A and Group C	Group B and Group C
**Post-VAS**	Mean difference	−0.27	−1.62	−1.34
	95% CI	−1.31–0.76	−2.65–−0.58	−2.38–−0.30
	*p*-value	1.00	0.001 *	0.007 *
**Post-Vit D**	Mean difference	4.03	6.33	2.29
	95% CI	1.27–6.78	1.53–7.04	−2.50–3.00
	*p*-value	0.002 *	0.001 *	0.02
**Post-PCS**	Mean difference	6.00	4.33	1.67
	95% CI	2.15–9.85	0.48–8.18	−5.52–2.18
	*p*-value	0.001 *	0.023 *	0.860
**Post-MCS**	Mean difference	5.20	3.67	1.53
	95% CI	2.05–8.35	0.52–6.82	−4.68–1.62
	*p*-value	0.001 *	0.018 *	0.695
**Post-12MWT**	Mean difference	153.07	70.07	83.00
	95% CI	65.30–240.83	−17.70–157.83	4.76–170.76
	*p*-value	0.0001 *	0.159	0.069

SD: standard deviation *p*-value: probability value * Significant (*p* < 0.05).

**Table 4 nutrients-13-01819-t004:** The relationship between the groups and the level of adherence.

	Adherence			
Group of the Study		Adherent	Non-Adherent	Total
Group A	Number of patients	12	3	15
	Expected Number	10.33	4.67	15.0
	% within Group	80%	20%	100%
Group B	Number of patients	10	5	15
	Expected Number	10.33	4.67	15
	% within Group	66.67%	33.33%	100%
Group C	Number of patients	9	6	15
	Expected Number	10.33	4.67	15
	% within Group	60%	40%	100%
Total	Number of patients	31	14	45
	Expected Number	31.0	14.0	45.0
	% within Groups	68.9%	31.1%	100%

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
