# Peer review of "Efficacy of Vitamin D Supplementation in Addition to Aerobic Exercise Training in Obese Women with Perceived Myalgia: A Single-Blinded Randomized Controlled Clinical Trial"

_nutrients, 2021, doi:10.3390/nu13061819_

Round 1

Reviewer 1 Report

The aim of this manuscript was to determine the impact of vitamin D (VitD) supplementation in combination with aerobic training on obese women with myalgia. The authors undertook a single blinded parallel study design whereby participants received either VitD supplementation and 12 weeks aerobic training, VitD supplementation alone, or aerobic training. A combination of self-reported measures to evaluate perceived myalgia, quality of life and walking distance. Additionally, all participants received a calorie balanced diet throughout.

There is a growing body of interest in the impact of VitD supplementation and exercise interventions, with limited data regarding their combination upon health related outcomes. While the data is of potential interest the English language requires extensive editing making interpretation of the authors justification and discussion difficult. Important factors require further discussion and investigation.

Major comments:

- Can the authors fully detail the justification and composition of the low calorie balanced diet? This is especially of importance as data presented clearly demonstrates all three groups had a reduction in BMI over the intervention period, which seemingly was independent of aerobic training.

- Why was the VitD supplementation dosage chosen to be 50,000IU per week? The reference given is of a meta-analysis and so is unclear. Additionally, could the authors discuss if serum calcium was measured to assess associated hypercalcemia?

- Could the authors fully discuss the observation of increased VitD levels in group C (aerobic training)? As VitD is stored within adipose tissue, decreased adiposity may result in the release of VitD into blood circulation and the increased observed.

- The authors specify outliers were removed from analysis. Can the authors please how these were determined and how many were removed?

Minor comments:

Abstract – The authors should state that participants were on calorie deficient diets

- Line 44 - The VitD abbreviation isn’t specified

- Line 88 - Could the authors specify why they chose a BMI of 30.34.9 kg/m2?

- Line 106 – The authors specify 20 participants were required for each group, however only 15 were recruited. Could the authors please discuss why this was chosen?

- Line 120 – What time of year were participants recruited and were venous blood samples taken in a fasted state?

- Line 192 – Could the authors specify the supplier of the Vitamin D capsules

- Line 348 – The referenced study is a systematic review, this may be an incorrect citation.

- Line 352 – The diet used and other diets are not discussed in the manuscript and should be expanded upon.

- Lines 359-362 – The meaning of the conclusion is unclear.

Author Response

Letter to the editor

Response to the reviewers' suggestions

Thank you for the comments that improved our manuscript. The following are the responses to all comments made by the reviewer:1

Regarding the extensive language editing, the full manuscript was revised by American Manuscript Editors  agency.  

Any changes in the manuscript were done in yellow color. 

Reviewer: 1

1- Can the authors fully detail the justification and composition of the low-calorie balanced diet? This is especially of importance as data presented clearly demonstrates all three groups had a reduction in BMI over the intervention period, which seemingly was independent of aerobic training.

Response *detail the justification of low-calorie balanced diet was presented in introduction section in yellow color as illustrated in the following manner:  

Obesity considers to be the main risk factor that led to insulin resistance, impaired glucose tolerance, dyslipidemia, dysfunctional bleeding, cardiovascular diseases as well as increasing the risk factor for developing the Poly Cystic Ovary (PCO) [ I]. Several studies had focused on the significant relation between pain and obesity, also on the effect of obesity on soft‐tissue structures, such as muscle, tendons, fascia, and cartilage [II, III, IV]

  1. Scott M. Grundy, Obesity, Metabolic Syndrome, and Cardiovascular Disease, The Journal of Clinical Endocrinology & Metabolism. 2004; 89(6): 2595–2600.
  2. Riddle DL, Pulisic M, Pidcoe P, Johnson RE. Risk factors for plantar fasciitis: a matched case‐control study. J Bone Joint Surg 2003; 85A: 872–877.
  • Ding C, Cicuttini F, Scott F, Cooley H, Jones G. Knee structural alteration and BMI: a cross‐sectional study. Obes Res 2005; 13: 350–361
  1. Wendelboe AM, Hegmann KT, Gren LH, Alder SC, White GL Jr, Lyon JL. Associations between body‐mass index and surgery for rotator cuff tendinitis. J Bone Joint Surg Am 2004; 86‐A: 743–747.

Weight loss could reduce the risk factors for type 2 diabetes, cardiovascular disease, endocrine and reproductive parameters in obese female subjects [A], also help obese chronic pain patients as well [B].

Therefore, lifestyle factors, as well as dietary intakes and physical activity that target weight reduction, have been demonstrated to affect the pain condition in obese subjects [A, B]. It is well established that exercise and/or engaging in hypocaloric diet [C, D, A] improves body composition in overweight and obese individuals by reducing systemic insulin levels which, in turn, reduces adipogenicity and lipogenic mechanisms [E, F]. Beyond diet and exercise interventions, supplementing the diet with vitamin D could enhance the weight loss and modify the metabolic alterations associated with obesity, could modify obesity-disease relations [G, H].

also many studies  revealed that the aerobic exercises had moderate beneficial effects on physical function , tender points and pain in patients with  fibromyalgia [F], also the effect of exercise accompanied to vitamin D3 supplementation on improvement of muscle strength in elderly adults was approved [G] , there was a incompatible proof encompassing the effectiveness of vitamin D supplementation alone or in combination with exercise on obese patients with fibromyalgia in addition to hypocaloric balanced diets [H].

  1. Panidis D, Farmakiotis D, Rousso D, Kourtis A, Katsikis I, Krassas G . Obesity, weight loss, and the polycystic ovary syndrome: effect of treatment with diet and orlistat for 24 weeks on insulin resistance and androgen levels. Fertil Steril.2008; 89(4):899–906
  2. Khoueir P, Black MH, Crookes PF, Kaufman HS, Katkhouda N, Wang MY. Prospective assessment of axial back pain symptoms before and after bariatric weight reduction surgery. Spine J. 2009;9(6): 454–463.
  3. Johansson, K.; Neovius, M.; Hemmingsson, E. E_ects of anti-obesity drugs, diet, and exercise on weight-loss maintenance after a very-low-calorie diet or low-calorie diet: A systematic review and meta-analysis of randomized controlled trials. Am. J. Clin. Nutr. 2014, 99, 14–23. [CrossRef]
  4. Poirier, P.; Despres, J.P. Exercise in weight management of obesity. Cardiol. Clin. 2001, 19, 459–470. [CrossRef]
  5. Arsenault, B.J.; Cote, M.; Cartier, A.; Lemieux, I.; Despres, J.P.; Ross, R.; Earnest, C.P.; Blair, S.N.; Church, T.S. Effect of exercise training on cardiometabolic risk markers among sedentary, but metabolically healthy overweight or obese post-menopausal women with elevated blood pressure. Atherosclerosis 2009, 207,530–533.
  6. Gordon, M.M.; Bopp, M.J.; Easter, L.; Miller, G.D.; Lyles, M.F.; Houston, D.K.; Nicklas, B.J.; Kritchevsky, S.B. Effects of dietary protein on the composition of weight loss in post-menopausal women. J. Nutr. Health Aging 2008, 12, 505–509.
  7. Kerksick CM, Roberts MD, Campbell BI, Galbreath MM, Taylor LW, Wilborn CD, Lee A, Dove J, Bunn JW, Rasmussen CJ, Kreider RB. Differential Impact of Calcium and Vitamin D on Body Composition Changes in Post-Menopausal Women Following a Restricted Energy Diet and Exercise Program. Nutrients. 2020 Mar 7;12(3):713. doi: 10.3390/nu12030713. PMID: 32156010; PMCID: PMC7146554.
  8. Mason C, Xiao L, Imayama I, Duggan C, Wang C-Y, Korde L, McTiernan A. Vitamin D3supplementation during weight loss: a double-blind randomized controlled trial. Am J Clin Nutr 2014;99:1015–25.

- Also, the effect of well-balanced low-calorie diet was well illustrated in discussion section in the following manner in yellow color:

The current study results showed significant improvement in the BMI in the three groups, with more improvement in group A, this improvement in all groups of the study could be mainly linked to the effect of low calorie balanced diet, that had positive effect alone on the weight and BMI [old reference, A] , also the more positive effect in group A could be linked to the effect of aerobic exercise as approved by many previous studies[old references] as well as due to the effect of Vitamin D supplementation in addition to aerobic exercise as illustrated in previous study which reported that when vitamin D intake is combined

with exercise training, resulted in more improvements in body composition, abdominal fat, blood lipid, and insulin resistance index than situations with one single treatment [B]. Also the current study result revealed that group B had the lowest improvement in BMI, as the improvement in this group might be due to the effect of balanced low calorie diet only while the Vitamin D supplementation alone had no significant effect on BMI, this was in line by the recent systemic review result which concluded that despite increased serum 25(OH)D levels, the obesity indices (BMI, Waist Circumference and Waist hip Ratio) did not improve significantly. [C].

  1. Suyardi MA, Johanes W, Harahap IP. The effects of balanced low-calorie diet on body composition and serum leptin of obese women. Med J Indones . 2005;14(4):220-4.
  2. Kim HJ, Kang CK, Park H, Lee MG. Effects of vitamin D supplementation and circuit training on indices of obesity and insulin resistance in T2D and vitamin D deficient elderly women. J Exerc Nutrition Biochem. 2014 Sep;18(3):249-57.
  3. L, Han L, Liu Q, Zhao Y, Wang L, Wang Y: Effects of Vitamin D Supplementation on General and Central Obesity: Results from 20 Randomized Controlled Trials Involving Apparently Healthy Populations. Ann Nutr Metab 2020;76:153-164. doi: 10.1159/000507418

**detail of the composition of low-calorie balanced diet was presented in intervention section (low-calorie balanced diet) in yellow color as illustrated in the following manner:

Low calorie balanced diet was designed based on in personal characteristics of each subject and with the goal of 500 Kcal daily caloric restriction than the total energy requirements (TEE) that calculated by Mifflin formula [A]. This prescribed individualized regimen was designed by the nutritionist every 14 days. A well-balanced low-calorie diet program, that involved 3 meals was designed according to the recommended dietary intake (RDI), while the macronutrients were considered in the diet, as follows: fat 20 ~ 30%, protein 10 ~ 15% and carbohydrates 55 ~ 65% of TEE, and rich in dietary fiber, as well as fruit and vegetables [A, old reference].

  1. Lotfi-Dizaji L, Mahboob S, Aliashrafi S, Vaghef-Mehrabany E, Ebrahimi-Mameghani M, Morovati A. Effect of vitamin D supplementation along with weight loss diet on meta-inflammation and fat mass in obese subjects with vitamin D deficiency: A double-blind placebo-controlled randomized clinical trial. Clin Endocrinol (Oxf). 2019 Jan;90(1):94-101.

2 - Why was the VitD supplementation dosage chosen to be 50,000IU per week? The reference given is of a meta-analysis and so is unclear.

Response: the dose was chosen to be 50,000 IU per week according to recent Randomized control trial [ A ]  in addition to the meta-analysis  that was mentioned previously in the text of manuscript in the following manner :

Vitamin D supplementation: It was taken by subjects in groups (A and B). One capsule containing cholecalciferol 50000 IU; (Biodal® 50 000 IU cholecalciferol; HAYAT PHARMACEUTICAL INDUSTRIES, Amman, Jordan) was taken every week [A].

ِA. Lotfi-Dizaji L, Mahboob S, Aliashrafi S, Vaghef-Mehrabany E, Ebrahimi-Mameghani M, Morovati A. Effect of vitamin D supplementation along with weight loss diet on meta-inflammation and fat mass in obese subjects with vitamin D deficiency: A double-blind placebo-controlled randomized clinical trial. Clin Endocrinol (Oxf). 2019 Jan;90(1):94-101.

Additionally, could the authors discuss if serum calcium were measured to assess associated hypercalcemia?

Response: in the current study the authors aimed to investigate the combined effects of aerobic interval training and vitamin D supplementation on functional capacity and perceived myalgia in middle aged obese women.

According to Shah and Chauhan 2016 who concluded that Obesity is certainly a factor for developing Vitamin D Deficiency and secondary hypercalcemia, also it could be possible to decrease morbidity in obese individuals by improving Vitamin D status only [ A]. Another recent study reported that calcium appeared to have no added benefit to dietary and exercise effects on body composition [B].

  1. Shah P, P Chauhan A. Impact of obesity on vitamin D and calcium status. Int J Med Res Rev. 2016Feb; 4(2):273-80.
  2. Kerksick CM, Roberts MD, Campbell BI, Galbreath MM, Taylor LW, Wilborn CD, Lee A, Dove J, Bunn JW, Rasmussen CJ, Kreider RB. Differential Impact of Calcium and Vitamin D on Body Composition Changes in Post-Menopausal Women Following a Restricted Energy Diet and Exercise Program. Nutrients. 2020 Mar 7;12(3):713. doi: 10.3390/nu12030713. PMID: 32156010; PMCID: PMC7146554.

Moreover, the assessment of secondary hypercalcemia due to obesity could be considered as a limitation in the present study as shown in limitation section in yellow color, but it still out of the main aim of the study.

Moreover, the present study did not examine the secondary hypercalcemia resulted from obesity, so future studies are warranted to investigate the effect of adding different types of exercises to Vitamin D supplementation with calcium level monitoring.

  3-Could the authors fully discuss the observation of increased VitD levels in group C (aerobic training)? As VitD is stored within adipose tissue, decreased adiposity may result in the release of VitD into blood circulation and the increased observed.

Response:

Improving vitamin D status with modest lifestyle modifications was recently suggested. The National Health and Nutrition Examination Survey (NHANES III) reports indicated that physical activity is related to serum 25(OH)D3 either due to enhanced vitamin D metabolism or increased sun exposure [A]. Also, The aerobic exercise was effective in improving vitamin D status in diabetic rats resulting in significantly higher serum vitamin D levels associated with increased vitamin D receptors in muscle, pancreas and adipose tissue. The obtained inverse correlations between vitamin D levels with serum glucose, insulin and HOMA indicate that moderate exercise could enhance vitamin D formation which in turn increases insulin sensitivity, reduce glucose and insulin levels [B].

  1. Wanner M, Richard A, Martin B, Linseisen J, Rohrmann S. Associations between objective and self-reported physical activity and vitamin D serum levels in the US population. Cancer Causes Control 2015;26:881–91.
  2. Aly YE, Abdou AS, Rashad MM, Nassef MM. Effect of exercise on serum vitamin D and tissue vitamin D receptors in experimentally induced type 2 Diabetes Mellitus. J Adv Res. 2016;7(5):671-679. doi:10.1016/j.jare.2016.07.001

This was well illustrated in the discussion section in yellow color in the following manner:

The results of the present study revealed that there was a significant increase in serum vitamin D in all groups with more significant increase in group A than either of B and C, the maximum  improvement in group A then in group B could be due to the effect of vitamin D supplementation as reported by Pilz et al.,[old references] who find out positive effects of vitamin D supplementation on serum 25-Hydroxyvitamin D concentrations , while the improvement of serum vitamin D in group C which received the aerobic exercise only could be correlated with increased vitamin D receptors in muscle, pancreas and adipose tissue [B],  this results could be supported by the results of Gholamalizadeh et al.[old reference] who concluded that serum vitamin D level increased significantly at the end of the training program in overweight and obese adolescents, also the current result was in alinement was previous report of The National Health and Nutrition Examination Survey (NHANES III) which reported that physical activity could positively affect the serum 25(OH)D3 due to enhanced vitamin D metabolism [A]

  1. The authors specify outliers were removed from analysis. Can the authors please how these were determined and how many were removed?

Response: Outliers wast detected by box and whiskers plots. Where the outlier is defined as a data point that is located outside the whiskers of the box plot[A]

outliers from pre treatment data of 1 subject in Group C were lost during the follow up process,   so  statistical analysis in this study was conducted according to intention to treat analysis in which multiple imputation model was used for the missing data from outlier removal box and whiskers plot. This technique allows for the uncertainty about the missing data by creating several different plausible imputed data sets and appropriately combining results obtained from each of them [B].

  1. McLeod, S. A. What does a box plot tell you? Simply psychology. 2019; July 19.
  2. Sterne JAC, White IR, Carlin JB, Spratt M, Royston P, Kenward MG, Wood AM, Carpenter JR: Multiple imputation for missing data in epidemiological and clinical research: potential and pitfalls. BMJ. 2009;338:157–60.

This was illustrated in statistical analysis section in yellow color as well as in Figure 1. that demonstrated the flow chart in accordance with the CONSORT statement.

Minor comments:

1.Abstract – The authors should state that participants were on calorie deficient diets.

Response: the abstract was updated in yellow color

Abstract:  Obese women were more susceptible for myalgia due to significantly lower vitamin D concentrations; present study investigated the efficacy of Vit D in addition to an aerobic interval in management of obese women with myalgia. 45 obese women had vitamin D deficiency myalgia; their age ranged from 30 to 40 years were assigned randomly into 3 equal groups. Group (A) received an aerobic interval training with vitamin D supplementation, while Group (B) received vitamin D supplementation only and Group (C) received an aerobic interval training only, participants on all groups were on calorie deficient diets. The study outcomes were Visual Analogue Scale (VAS) for pain evaluation, Serum vitamin D level, Cooper 12-Minute Walk Test for functional capacity evaluation while short form health survey (SF) was used for assessment of quality of life. This study results revealed a significant improvement of pain intensity level, serum vitamin D level and quality of life in all groups with significant difference between group A and either of (B and C), also revealed a significant improvement in functional capacity in group A and C with no significant change in group (B). Aerobic interval training with vitamin D supplementation were more effective for management of obese women with perceived myalgia.

- Line 44 - The VitD abbreviation isn’t specified

Response : it was corrected in the following manner in yellow color

Reduction of Vitamin D (3) bioavailability from dietary and cutaneous sources is the main couse of vitamin D insufficiency associated with obesity due to the deposition in body fat compartments [2].

Line 88 - Could the authors specify why they chose a BMI of 30.34.9 kg/m2?

Response : the author select a first class of obesity according to the following reference , and the justification was done by adding the class of obesity and its reference to the text in yellow color

Participants: Sixty obese women suffered from nonspecific muscle pain that referred to physical therapy by a specialized orthopedist. The patients were enrolled to this study if they were with age range of 30 and 40 years, their body mass index (BMI) ranging in  class I obesity (BMI, 30.0 to 34.9 kg/m2) [A]

  1. Aronne, L.J. (2002), Classification of Obesity and Assessment of Obesity‐Related Health Risks. Obesity Research, 10: 105S-115S.

- Line 106 – The authors specify 20 participants were required for each group, however only 15 were recruited. Could the authors please discuss why this was chosen?

Response: according to previous studies, sample size calculation demonstrated that the required participants was 13 subjects in each group, but the authors preferred to involve 20 subjects to avoid subject’s  loss from exclusion criteria. The thing that was occurred and 15 subjects were excluded, so the remaining 45 subjects were divided into the three groups to be 15 subjects in each group and still more than 13 that was required from previous studies.

This was illustrated in the following manner in yellow color.

According to previous studies, 13 patients in each group was estimated to be the minimum sample size which achieve a power of 90% to detect an effect size of 0.87 in the outcome variables, assuming a type II error (Alfa = 0.05, For considering the drop out during the study, the sample size was increased by 15% to be 20 patients in each group, to avoid the absence of enrolled subjects because of exclusion criteria. The remaining forty-five women with nonspecific muscle pain had vitamin D deficiency participated in the study and randomly assigned to three groups, (Group A = 15) received a combination of an aerobic interval training exercise program and vitamin D supplementation (cholecalciferol 50000 IU/week) [A]. (Group B=15) received vitamin D supplementation only (cholecalciferol 50000 IU/week), while (Group C=15): received an interval training exercise program only, while each participant in the three group followed an individualized low fat balanced diet.

  1. Lotfi-Dizaji L, Mahboob S, Aliashrafi S, Vaghef-Mehrabany E, Ebrahimi-Mameghani M, Morovati A. Effect of vitamin D supplementation along with weight loss diet on meta-inflammation and fat mass in obese subjects with vitamin D deficiency: A double-blind placebo-controlled randomized clinical trial. Clin Endocrinol (Oxf). 2019 Jan;90(1):94-101.

 Line 120 – What time of year were participants recruited and were venous blood samples taken in a fasted state?

What time of year were participants recruited?

Response: This was illustrated in participants section in the following manner in yellow color 

The study was conducted in outpatient clinic of the faculty of physical therapy Cairo University with first recruitment date in January 2020 until May 2020. 

were venous blood samples taken in a fasted state:

Response: Fasting (12 hours) venous blood sample (10 mL) was collected, and this was updated in laboratory analysis section in following manner in yellow color:

  1. Laboratory analysis: BioPlex ® 2200 System (Hitachi, model 704,902, Japan) was used for measuring serum vitamin D level (25OHD), Fasting (12 hours) venous blood samples (10 mL) were collected in the morning at 9 AM by a specialist technician from antecubital vein in Cairo University hospital clinical lab using an anticoagulant free tube (EDTA K3) [14].

- Line 192 – Could the authors specify the supplier of the Vitamin D capsules

Respons:It was updated in intervention section in following manner:

Vitamin D supplementation: It was taken by subjects in groups (A and B). One capsule containing cholecalciferol 50000 IU; (Biodal® 50 000 IU cholecalciferol; HAYAT PHARMACEUTICAL INDUSTRIES, Amman, Jordan) was taken every week [11, A].

  1. Lotfi-Dizaji L, Mahboob S, Aliashrafi S, Vaghef-Mehrabany E, Ebrahimi-Mameghani M, Morovati A. Effect of vitamin D supplementation along with weight loss diet on meta-inflammation and fat mass in obese subjects with vitamin D deficiency: A double-blind placebo-controlled randomized clinical trial. Clin Endocrinol (Oxf). 2019 Jan;90(1):94-101.

- Line 348 – The referenced study is a systematic review; this may be an incorrect citation.

Response: The requested correction was done, and the reference was changed by another one in the following manner:

The results of the present study contradicted with the finding of Westra et al. [A] who reported that Six months of vitamin D supplementation did not improve health-related quality of life (using the SF-36). The contradiction between the present and Westra et al. study may be due to the difference in the inclusive criteria of participants in each study, as this study was performed on non-vitamin D-deficient people with type 2 DM.

  1. Westra S, Krul-Poel YH, van Wijland HJ, Ter Wee MM, Stam F, Lips P, Pouwer F, Simsek S (2016) Effect of vitamin D supplementation on health status in non-vitamin D deficient people with type 2 diabetes mellitus. Endocr Connect 5(6):61–69.

- Line 352 – The diet used, and other diets are not discussed in the manuscript and should be expanded upon.

Response : detail of the composition of low-calorie balanced diet was presented in intervention section (low-calorie balanced diet) in yellow color as illustrated in the following manner:

Low calorie balanced diet was designed based on in personal characteristics of each subject and with the goal of 500 Kcal daily caloric restriction than the total energy requirements (TEE) that calculated by Mifflin formula [A]. This prescribed individualized regimen was designed by the nutritionist every 14 days. A well-balanced low-calorie diet program, that involved 3 meals was designed according to the recommended dietary intake (RDI), while the macronutrients were considered in the diet, as follows: fat 20 ~ 30%, protein 10 ~ 15% and carbohydrates 55 ~ 65% of TEE, and rich in dietary fiber, as well as fruit and vegetables [A,28].

  1. A. Lotfi-Dizaji L, Mahboob S, Aliashrafi S, Vaghef-Mehrabany E, Ebrahimi-Mameghani M, Morovati A. Effect of vitamin D supplementation along with weight loss diet on meta-inflammation and fat mass in obese subjects with vitamin D deficiency: A double-blind placebo-controlled randomized clinical trial. Clin Endocrinol (Oxf). 2019 Jan;90(1):94-101.

And in discussion section:

The current study had some limitation that could be considered. One of this limitation was the investigation of the effect of adding the aerobic interval exercise to vitamin D supplementation with low balanced macronutrient diet only without true control or placebo group. So, the results of the present study should be confirmed by other studies with different types of diet as high protein diet or Mediterranean diet with using true control or placebo group.

- Lines 359-362 – The meaning of the conclusion is unclear.

Response: The conclusion section was updated in the following manner:

Vitamin D supplementation combined with aerobic exercise in addition to well-balanced hypo caloric diet could provide a novel protocol to improve the non-specific musculoskeletal pain, functional capacity, and the quality of life in Obese Female with Perceived Myalgia. 

Reviewer 2 Report

Current paper analyses the efficacy of Vitamin D supplementation after aerobic exercise in obese females with perceived myalgia. The authors included 45 women. The studied population was equally divided them in 3 groups: Group A: aerobic training + Vitamin D supl., Group B: Vitamin D supl. and Group C: aerobic training

The paper is well written. The used English is very good. The statistical analysis used in the paper is adequate to elucidate differences between the groups. The data are well presented.

I do have some methodological concerns about the paper: A control (or placebo) group is missing. Maybe the authors can include a placebo group in their study (e.g. Group D without any intervention). If this is not possible, maybe the authors can explain why they did not include a placebo-group and what data would be expected for the read-out parameters of the study, for the placebo group.

Additionally, I would encourage the authors to include a section in the “Discussion” were pathophysiological action of aerobic training and vitamin D supplementation is explained especially on obese patients with perceived myalgia. => How does vitamin D and aerobic training work on this patients?

Author Response

Letter to the editor

Response to the reviewers' suggestions

Reviewer 2:

Thank you for the comments that improved our manuscript. The following are the responses to all comments made by the reviewer: 2

  1. I do have some methodological concerns about the paper: A control (or placebo) group is missing. Maybe the authors can include a placebo group in their study (e.g. Group D without any intervention). If this is not possible, maybe the authors can explain why they did not include a placebo-group and what data would be expected for the read-out parameters of the study, for the placebo group.

Response:

The use of placebo controls in clinical trials remains controversial. Ethical analysis and international ethical guidance permit the use of placebo controls in randomized trials when scientifically indicated in four cases: (1) when there is no proven effective treatment for the condition under study; (2) when withholding treatment poses negligible risks to participants[A] The main aim of this study was to investigate the effect of Vitamin D Supplementation in Addition to Aerobic Exercise Training in Obese Female with Perceived Myalgia, so from the previous introduction in this manuscript the Obese Female with Perceived Myalgia condition does not consider a condition that does not have proven effective treatment[B}.  Also, according to McKillip 1992[C], the benefits and costs of control groups shift as research is moved from laboratory to field settings. In field research, experimental and control groups are more likely to differ systematically on important dimensions other than independent variables. Comparisons between groups may not yield the unbiased estimates of impact that accompany controlled laboratory research. At the same time, logistic considerations affect the cost of each observation more dramatically in field settings than in a laboratory. If gathering control group information requires additional data collection sites, costs of individual control observations may be significantly higher than costs of additional experimental group observations. Possibilities of nonequivalence and of increased costs suggest a need for unbiased research designs that do not require control groups[C]. In the present study the  authors used a sound research design to compare the isolated effect of either the Vitamin D supplementation or the aerobic exercise alone and the combined effect of both of them, without using a true control group as if the authors included control group, they should include control group for each of aerobic exercise and vitamin D supplementation so holding one treatment  on one group, to avoid presentation of one group didn’t receive any type of intervention.  

Also, according to your valuable comment, it could be another limitation of present study that didn’t involve a true control group as illustrated in limitation section in yellow color:

The current study had some limitation that could be considered. One of this limitation was the investigation of the effect of adding the aerobic interval exercise to vitamin D supplementation with low balanced macronutrient diet only without true control or placebo group. So, the results of the present study should be confirmed by other studies with different types of diet as high protein diet or Mediterranean diet with using true control or placebo group.

  1. Millum J, Grady C. The ethics of placebo-controlled trials: methodological justifications. Contemp Clin Trials. 2013;36(2):510-514. doi:10.1016/j.cct.2013.09.003
  2. McVinnie DS. Obesity and pain. Br J Pain. 2013;7(4):163-170. doi:10.1177/2049463713484296
  3. McKillip J. (1992) Research without Control Groups. In: Bryant F.B. et al. (eds) Methodological Issues in Applied Social Psychology. Social Psychological Applications to Social Issues, vol 2. Springer, Boston, MA. https://doi.org/10.1007/978-1-4899-2308-0_8

Additionally, I would encourage the authors to include a section in the “Discussion” were pathophysiological action of aerobic training and vitamin D supplementation is explained especially on obese patients with perceived myalgia. => How does vitamin D and aerobic training work on this patients?

Response: the required underlying pathophysiology was added to discussion section in the following manner in yellow color:

This improvement of pain level in group A, and group B could be because of improvement of Vitamin D level as well as to aerobic exercise, as the Improvement in vitamin D status in obese subjects with vitamin D deficiency resulted in weight, fat mass and monocyte chemoattractant protein‐1(MCP‐1) decrease. Weight loss and vitamin D supplementation may act synergistically to reduce levels of meta‐inflammation [A].

  1. Lotfi-Dizaji L, Mahboob S, Aliashrafi S, Vaghef-Mehrabany E, Ebrahimi-Mameghani M, Morovati A. Effect of vitamin D supplementation along with weight loss diet on meta-inflammation and fat mass in obese subjects with vitamin D deficiency: A double-blind placebo-controlled randomized clinical trial. Clin Endocrinol (Oxf). 2019 Jan;90(1):94-101. doi: 10.1111/cen.13861. Epub 2018 Nov 12. PMID: 30246883.

Round 2

Reviewer 1 Report

The authors have robustly addressed all my comments raised.